# Coral fluorescence: a prey-lure in deep habitats

Or Ben-Zvi [1,2✉], Yoav Lindemann [2,3], Gal Eyal [1,2,4,5] & Yossi Loya[1]

Fluorescence is highly prevalent in reef-building corals, nevertheless its biological role is still under ongoing debate. This feature of corals was previously suggested to primarily screen harmful radiation or facilitate coral photosynthesis. In mesophotic coral ecosystems (MCEs; 30-150 m depth) corals experience a limited, blue-shifted light environment. Consequently, in contrast to their shallow conspecifics, they might not be able to rely on photosynthates from their photosymbionts as their main energy source. Here, we experimentally test an alternative hypothesis for coral fluorescence: a prey-lure mechanism for plankton. We show that plankton exhibit preferential swimming towards green fluorescent cues and that compared to other morphs, higher predation rates are recorded in a green fluorescing morph of the mesophotic coral *Euphyllia paradivisa*. The evidence provided here - that plankton are actively attracted to fluorescent signals - indicates the significant role of fluorescence in amplifying the nutritional sink adjacent to coral reefs.

[1] School of Zoology, Faculty of Life Sciences, Tel Aviv University, Tel Aviv-Yafo, Israel. [2] The Interuniversity Institute for Marine Sciences in Eilat, Eilat, Israel. [3] The Fredy & Nadine Herrmann Institute of Earth Sciences, The Hebrew University of Jerusalem, Jerusalem, Israel. [4] ARC Centre of Excellence for Coral Reef Studies, School of Biological Sciences, The University of Queensland, Brisbane, QLD, Australia. [5] The Mina & Everard Goodman Faculty of Life Sciences, Bar-Ilan University, Ramat Gan, Israel. ✉email: orbzvi@gmail.com

Coral fluorescence has long been studied in the marine environment. This striking feature of corals is attributed to the green fluorescent protein (GFP) family[1,2], which can convert light's wavelength. These proteins were first discovered in close relatives of stony corals[3] and are widely used as visual markers in many fields of research. Several hypotheses regarding the biological role of fluorescence in corals have been examined: photoprotection (i.e., the sunscreen hypothesis)[4–8], photosynthesis enhancement[9,10], antioxidant activity[11,12], protection against herbivory[13], and attraction of symbiotic algae to the corals[14,15]. Coral fluorescence displays a light-dependent regulation, upregulated mostly by the presence of blue light[16] or varying light intensities[7,8,16,17]. Furthermore, fluorescent proteins (FPs) will usually convert short energetic wavelengths into longer, less harmful, wavelengths, thereby supporting the photoprotection hypothesis.

Corals are believed to rely heavily on the photosynthates produced by their associated algal symbionts (family Symbiodiniaceae) as an energy source[18]. Consequently, some studies have suggested that coral fluorescence might improve or contribute to the photosynthesis performed by the symbiotic algae by broadening the light spectrum or enhancing the available light by reflecting it[19,20]. Such a role, however, is still controversial[21–24].

The role of fluorescence in mesophotic coral ecosystems (MCEs; 30–150 m depth)[25] has only recently started to gain scientific interest, despite MCEs hosting highly fluorescent corals and a high occurrence of fluorescence polymorphism[26]. When two of the major hypotheses for coral fluorescence were tested on mesophotic corals, no evidence was found for the enhancement of photosynthesis[27–29] or for the photoprotective role[29], hence, coral fluorescence in deeper habitats creates a gap in our understanding of the widespread phenomenon.

It is has been established that corals can display a dual nutritional strategy[30,31]. While autotrophy can exceed the coral energetic requirements, heterotrophy can fulfill up to 50% of the metabolic needs in healthy corals[32] and up to 100% during bleaching events[30]. Although coral trophic states can be species specific[33–35] and subjected to environmental conditions[34–36], it is assumed that autotrophy-based nutrition typifies shallow-water corals, where light is not a limiting factor. In contrast, MCEs are characterized by much lower light intensities, i.e., 1–10% of surface irradiance, and a narrower spectrum centered around the blue region of the spectrum[37], resulting in a possibly impaired ability to acquire energy through photosynthesis[38]. While there is evidence of mixed nutritional sources for corals along a depth gradient[38–40], some species do present a transition from photosynthesis-based nutrition (autotrophic) to predation-based nutrition (heterotrophic) in shallow versus deep corals[18,41–43]. Thus, it can be expected that corals inhabiting deeper reefs should possess prey-lure abilities and/or prey-capture mechanisms. Moreover, whereas most shallow corals contract their tentacles during the daytime and only expand them during the night[44] in order to avoid radiation-induced damage[45] and depending on other factors (i.e., flow rate, photosymbionts density, oxygen concentration, and presence of food)[44–46], some mesophotic coral species display fully extended tentacles during the day[29,47]. Tentacle extension and mouth opening, usually at nighttime, are the prime prerequisites enabling predation in corals and in mesophotic corals these conditions are also met during daytime. To date, the potential of fluorescence to serve as a mechanism supporting the heterotrophic needs of corals has only been hypothesized[48] but not experimentally tested. Here, we suggest that fluorescence in corals constitutes a potential attractant for prey, especially in mesophotic environments.

## Results

**The attraction of plankton to fluorescence.** In order to examine the potential attraction of plankton to fluorescence, we chose the "lab rat" crustacean *Artemia salina*. The adult brine shrimp *A. salina* demonstrates good motility, both naupliar and compound eyes and, in our ex situ attraction experimental setup, it displayed preferential swimming as follows: when given a choice between a fluorescent target (fluorescent green or fluorescent orange) versus a clear target (i.e., "control") located on opposite sides of a chamber and illuminated by blue light (which excites fluorescence), *A. salina* showed a significant preference for the fluorescent target (Fig. 1a, b; GLMM, $p = 2e-16$, Cohen's $d = 4.13$ for fluorescent green and $p = 5.6e-07$, Cohen's $d = 2.03$ for fluorescent orange); when given a choice between two clear targets, *A. salina* were observed to be randomly distributed in the experimental setup (Fig. 1c; GLMM, $p = 0.88$, Cohen's $d = -0.17$); and when given a choice between a fluorescent green target and a fluorescent orange target, the fluorescent green target was significantly preferred (Fig. 1d; GLMM, $p = 0.002$, Cohen's $d = -1.44$). In order to determine whether the attraction was simply due to light reflectance or to light conversion (i.e., fluorescence), we tested a fluorescent target versus a reflective target with similar reflectivity. In both cases *A. salina* presented a preferential swimming toward the fluorescent target (Fig. 1e, f; GLMM, $p = 3.7e-07$, Cohen's $d = -2.09$ and $p = 1.4e-09$, Cohen's $d = -1.19$ for fluorescent green and fluorescent orange, respectively). We then tested a fluorescent target (fluorescent green and fluorescent orange) versus colored, non-fluorescent targets (green and orange). *A. salina* showed a preference for the fluorescent targets over the colored, non-fluorescent ones (Fig. 1g, h; GLMM, $p = 1.75e-07$, Cohen's $d = -1.52$ and $p = 5.8e-08$, Cohen's $d = -1.49$ for fluorescent green and fluorescent orange targets, respectively). Overall, in all our ex situ attraction experiments, *A. salina* displayed a preferential attraction toward a fluorescent cue.

Following the experiments with *A. salina*, using the same setup but on a smaller scale, we tested the attraction of *Anisomysis marisrubri* (commonly referred to as "mysids"), a native crustacean abundant in the Gulf of Eilat/Aqaba (GoE/A), northern Red Sea and representing a potential prey for corals, as well as the non-native *Sparus aurata* fish larvae, as an example of an organism not considered as coral prey. The mysids showed a preferential swimming toward a green fluorescent target over both a reflective target (Fig. 2a; GLMM, $p = 0.03$, Cohen's $d = -1$) and a non-fluorescent green target (Fig. 2e; GLMM, $p = 0.007$, Cohen's $d = -1$). However, the mysids displayed preferential swimming toward the reflective target over the orange fluorescent target (Fig. 2b; $p = 0.0009$, Cohen's $d = 1.92$) and random swimming in the presence of a non-fluorescent orange target (Fig. 2f; $p = 0.73$, Cohen's $d = 0.3$). When given the choice between two reflective targets, they exhibited random distribution between the two sides of the setup (Fig. 2c; GLMM, $p = 0.22$, Cohen's $d = 0.3$); and when given the choice between green or orange fluorescent targets, the mysids preferred the green fluorescent target (Fig. 2d; GLMM, $p = 1.9e-05$, Cohen's $d = -1.85$). The mysids thus showed active attraction to the green fluorescence while avoiding the orange fluorescence, and the orange color in general.

When the *S. aurata* fish larvae were given the choice between a fluorescent target versus a reflective target, they presented preferential swimming toward the reflective target (Fig. 3a, b; GLMM, $p = 6.9e-13$, Cohen's $d = -6.71$ and $p = 3.9e-07$, Cohen's $d = -5.52$ for green and orange fluorescent targets, respectively). As demonstrated both for *A. salina* and *A. marisrubri*, when presented with two reflective targets, the fish larvae did not demonstrate a preferential direction of swimming (Fig. 3c;

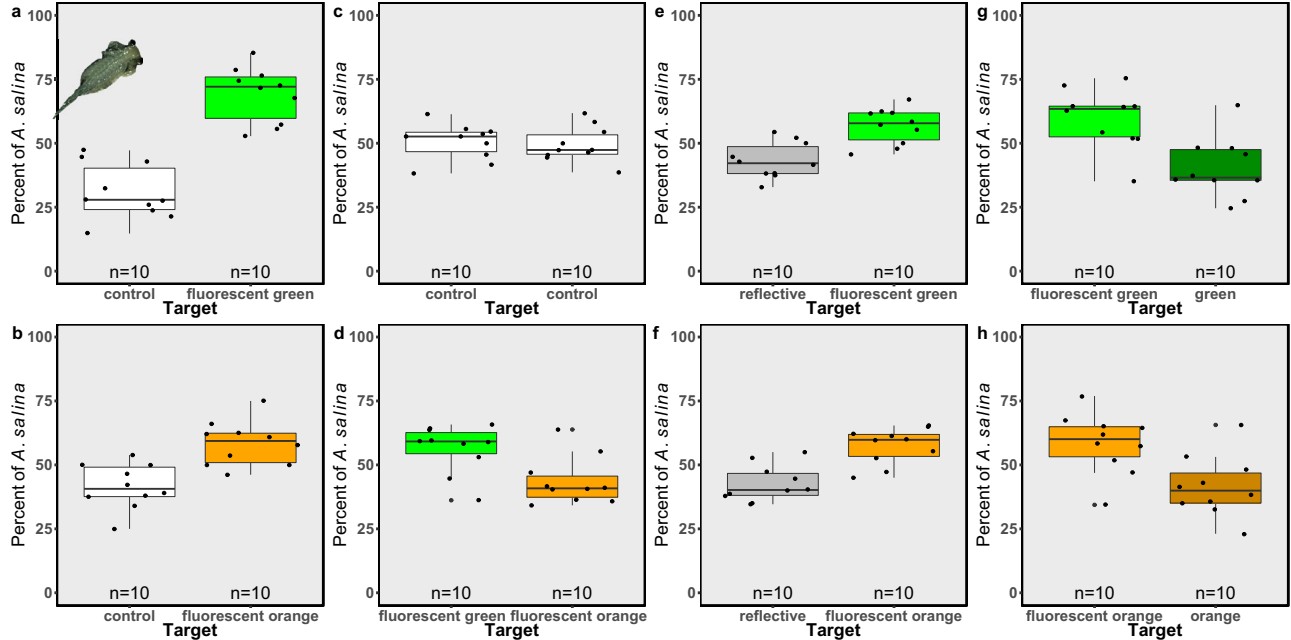

**Fig. 1 The attraction of *Artemia salina* to fluorescent and non-fluorescent targets.** Attraction to fluorescent (fluorescent green or fluorescent orange) or non-fluorescent (control, reflective, green, and orange) targets presented as the percentage of *A. salina* attracted to each target. **a** Clear target ("control"; white boxes) versus fluorescent green target (bright green boxes), **b** control target versus fluorescent orange target (bright orange boxes), **c** control target versus control target, **d** fluorescent green target versus fluorescent orange target, **e** fluorescent green target versus reflective target (gray boxes), **f** fluorescent orange target versus reflective target, **g** fluorescent green target versus non-fluorescent green target (dark green boxes), and **h** fluorescent orange target versus non-fluorescent orange target (dark orange boxes). $n = 10$ trials for each combination. Boxes represent the upper and lower quartile, centerlines represent medians, and whiskers extend to data measurements that are less than 1.5*IQR away from the first/third quartile.

GLMM, $p = 0.89$, Cohen's $d = 0.04$). The fish also did not show a preference when given a fluorescent green and non-fluorescent green target or between a fluorescent orange and non-fluorescent orange target (Fig. 3e, f; GLMM, $p = 0.68$, Cohen's $d = -0.27$ and $p = 0.29$, Cohen's $d = -1.59$ for green and orange, respectively).

We then performed an in situ experiment in order to examine the possible attraction of heterogenic, natural plankton assemblages to fluorescence in the sea, under natural currents and light conditions at 40 m depth, where fluorescence is naturally excited. We deployed three plankton traps, containing either an orange fluorescent, green fluorescent, or clear (i.e., "control") target (see Fig. 4a). The color of the trap was found to be significant, resulting in differences in the number of plankton attracted to it (Fig. 4b; ANOVA, $p = 0.03$) with the control trap attracting the lowest plankton concentration in all trials (Fig. 4b; LMM, $p = 0.02$, Cohen's $d = 1.88$ and $p = 0.02$, Cohen's $d = 1.68$ for green and orange, respectively). The average ± SD percentage of plankton was $40.2 ± 16.05\%$ in the green fluorescent trap, $37.6 ± 11.75\%$ in the fluorescent orange trap, and $22.15 ± 7.8\%$ in the control trap (Fig. 4b). The vast majority of plankton that actively entered the traps comprised the copepod family, together with some polychaetes, chaetognaths, bivalves, nematodes, and various crustaceans (detailed data provided in Supplementary Data 1).

**Predation success of *Euphyllia paradivisa* fluorescent morphs.**
To determine whether the fluorescence of corals contributes to their predation success, we used two fluorescence morphs of the mesophotic coral *Euphyllia paradivisa* (abundant in the GoE/A between 40–70 m depth[49] and recorded the number of *A. salina* captured by each morph during 30 min of predation in the laboratory. The green morph (Fig. 5a) presents a green fluorescent emission peak at 515 nm and a 0.52 fluorescence efficiency

(Fig. 5b), while the yellow morph (Fig. 5c) presents an emission peak at 545 nm and a 0.42 fluorescence efficiency (Fig. 5d).

Under blue light, which excites the natural coral fluorescence, the color morph of the coral had a significant effect on predation success (Fig. 6a; GLMM, $p = 0.01$, Cohen's $d = -0.46$, $n = 42$ trials for each morph), with the green morph demonstrating a higher predation success compared to the yellow morph. The green morph preyed upon $3.21 ± 1.4$ (mean ± SD) *A. salina* individuals per cm², while the yellow morph preyed upon $2.53 ± 1.04$ individuals. When the experiment was repeated under red illumination, which does not excite coral fluorescence, there was no significant difference between the predation success of the green and yellow morphs (Fig. 6b; LMM, $p = 1$, Cohen's $d = 1.47$, $n = 32$ trials for each morph, respectively).

**Discussion**
The underwater visual systems of marine organisms are extremely diverse and may differ among species, locations, and life stages. Sight, in most marine invertebrates (particularly crustaceans), is enabled through visual pigments that can collect light in the ultraviolet to red wavelengths (300 to <600 nm)[50]. For example, our tested crustacean *A. salina* presents a maximal positive phototaxis under low light intensities at 520 nm[51]. In Mysids, the maximal sensitivity can differ among species[52] and while the photosensitivity of *A. marisrubri* has not yet been characterized, measurements performed on other species revealed a phototactic response at wavelengths ranging between 395 to 540 nm[53,54]. Fish, specifically fish from habitats characterized by a more monochromatic light spectrum, will usually present with a wavelength-dependent photic sensitivity, stronger around the blue and green wavelengths[55,56]. Given the appropriate chromophores, certain planktonic taxa should be able to perceive fluorescent colors under the mesophotic light environment[57], possibly triggering a color-dependent phototaxis such as the

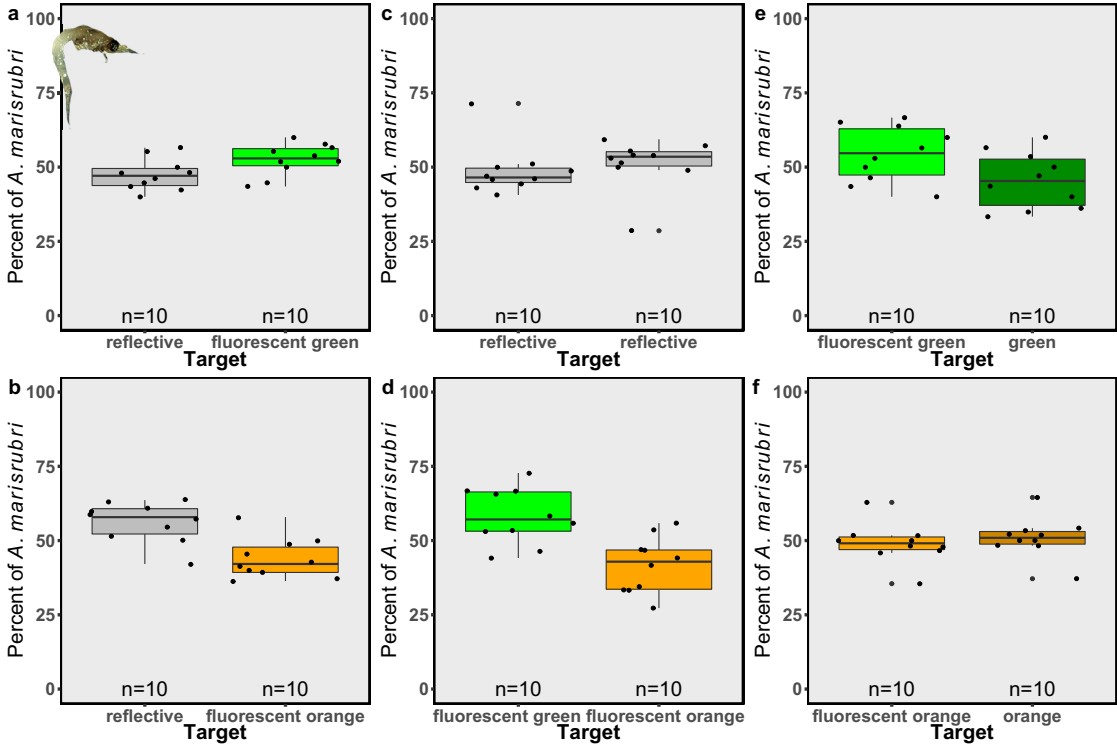

**Fig. 2 The attraction of *Anisomysis marisrubri* ("mysids") to fluorescent and non-fluorescent targets.** Attraction to fluorescent (fluorescent green or fluorescent orange) or non-fluorescent (reflective, green, and orange) targets presented as the percentage of *A. marisrubri* attracted to each target. **a** Reflective target (gray boxes) versus fluorescent green target (bright green boxes), **b** reflective target versus fluorescent orange target (bright orange boxes), **c** reflective target versus reflective target, **d** fluorescent green target versus fluorescent orange target, **e** fluorescent green target versus a non-fluorescent green target (dark green boxes), and **f** fluorescent orange target versus non-fluorescent orange target (dark orange boxes). $n = 10$ trials for each combination. Boxes represent the upper and lower quartile, centerlines represent medians, and whiskers extend to data measurements that are less than 1.5*IQR away from the first/third quartile.

behavior previously described in custaceans[58,59], bivalves[60], ciliates[61], and dinoflagellates[62]. The attraction, rather than avoidance, of plankton to bright green color in our experiments may be explained by a color-dependent behavior that was previously recorded in crustaceans and was suggested to be involved in food foraging[63]. As zooplankton prey on phytoplankton, the green color may be a positive stimulus for them, coinciding with the peak absorbance of visual pigments around greener wavelengths in many zooplankton species[64]. In other cases, when lacking the appropriate visual pigments, plankton may only be able to discriminate between the background and darker or lighter objects[50], which may also lead to a phototactic swimming response toward a fluorescent object that transmits an intensified signal compared to a non-fluorescent object. Accordingly, our attraction experiments revealed preferential swimming of plankton toward fluorescent rather than non-fluorescent objects, both in the laboratory and the sea. Bioluminescence in several marine organisms, e.g., angler fish[65], cephalopods[66,67], and siphonophores[68], was previously suggested as a trait used to lure prey. However, contrary to fluorescence, bioluminescence does not require an external light source for excitation, and the prey in these cases are usually equipped with developed eyes and a complex visual system. Moreover, because the above-noted predators are nocturnal, such a light source (i.e., bioluminescence) serves as a "light trap" against the dark background. Similarly, the "bait hypothesis" contends that the bioluminescence of marine bacteria, when consumed by transparent plankton, serves as an attractant for fish, which will favor predation on the "glowing" plankton and thereby increase the dispersal of the bacteria[69,70]. While in shallow habitats, fluorescence may not be as impressive

or easily visible as bioluminescence, the light environment found at mesophotic depths[24] provides a good excitation for most FPs while also enhancing the contrast between the surroundings and the colorful fluorescent signal. Furthermore, many corals are known to display a fluorescent color pattern that highlights their mouth or tentacle tips[62,71–73]. The combination of the excited FPs creating a visible stimulus and the fluorescent patterns highlighting specific parts of the corals supports the suggestion that fluorescence, similar to bioluminescence, acts as a prey attractant.

While the attraction of prey will increase the odds of a predation event, it will not ensure higher predation rates, as prey might escape or the predator might not consume the prey. Here, we have also demonstrated a higher plankton capture rate in a green fluorescent morph of a mesophotic coral. In its natural habitat in the mesophotic reefs of Eilat, the yellow morph of *E. paradivisa* was found to be the least abundant[29], which can now be potentially explained by the lower prey attraction to this color found in the present study (Fig. 6a). Although a prey-attraction role of fluorescence had been previously suggested, following the finding that fish juveniles were attracted to fluorescent signals[48], this role had not been explored to date, to the best of our knowledge, with respect to the important interaction between corals and plankton. We now provide evidence of such a relationship, demonstrating that coral fluorescence can act as a luring mechanism, actively attracting motile plankton prey to a sessile coral predator, particularly in habitats where corals require other sources of energy in addition to or instead of photosynthesis, and that a specific fluorescence emission can improve predation success. Moreover, from a broader perspective, it has been shown previously that plankton densities decline near the sea bottom in

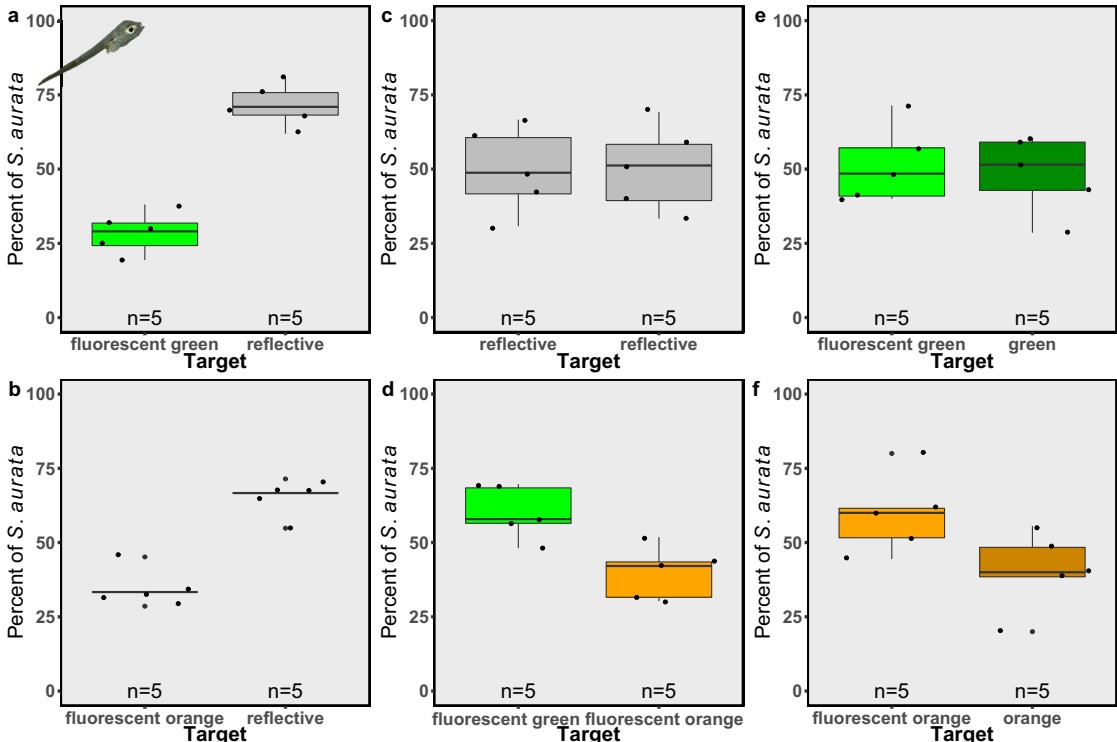

**Fig. 3 The attraction of *Sparus aurata* larvae to fluorescent and non-fluorescent targets.** Attraction to fluorescent (fluorescent green or fluorescent orange) or non-fluorescent (reflective, green, and orange) targets presented as the percentage of *S. aurata* attracted to each target. **a** Reflective target (gray boxes) versus fluorescent green target (bright green boxes), **b** reflective target versus fluorescent orange target (bright orange boxes), **c** reflective target versus reflective target, **d** fluorescent green target versus fluorescent orange target, **e** fluorescent green target versus a non-fluorescent green target (dark green boxes), and **f** fluorescent orange target versus non-fluorescent orange target (dark orange boxes). *n* = 5 trials for each combination. Boxes represent the upper and lower quartile, centerlines represent medians, and whiskers extend to data measurements that are less than 1.5*IQR away from the first/third quartile.

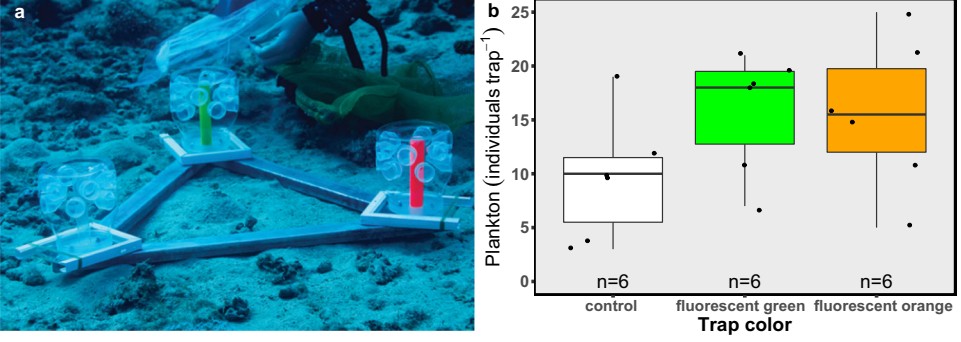

**Fig. 4 The attraction of natural plankton to fluorescent and non-fluorescent traps. a** The experimental setup for the in situ attraction experiment. **b** The results of the plankton attraction experiment are presented as the number of plankton individuals in each trap (*n* = 6 trials). Boxes represent the upper and lower quartile, centerlines represent medians, and whiskers extend to data measurements that are less than 1.5*IQR away from the first/third quartile.

coral-reef ecosystems. This near-bottom depletion effect has been attributed to the removal of plankton by suspension feeders[74], or to plankton avoidance of the benthic predators[75]. Our finding that plankton were actively attracted to fluorescent signals in both the traps and the live mesophotic corals, may indicate that plankton are being attracted to the benthic fluorescent predators and consumed by them. This latter point offers new insight into the causes behind this enhanced nutritional sink. Despite the gaps in our knowledge regarding the visual perception of fluorescence signals by plankton, we present experimental evidence for a prey-luring effect of fluorescence in corals. We suggest that this hypothesis, which we now term the "light trap hypothesis", may

also apply to other fluorescent organisms in the sea; and that this phenomenon may play a greater role in marine ecosystems than previously considered.

## Methods

To test the suggested prey-attraction role, we first determined whether plankton are attracted to fluorescence, in both the laboratory and the sea. We then quantified the predation abilities of mesophotic polymorphic fluorescent corals in the laboratory.

**Spectral analyses.** Targets and corals were spectrally analyzed for reflectance and fluorescence efficiency using a JAZ spectrophotometer system (Ocean Optics, USA) equipped with a 2 m long flat-cut optic fiber (2P600-2-UV-VIS). The reflectance of fluorescent and non-fluorescent targets was measured by directing the optic fiber at

either a >97% reflective standard (WS-1, Ocean Optics, USA) or at the target, illuminated by a blue collimated light (emission peak at 450 nm) at a 45° angle. Fluorescence efficiency of the fluorescent targets and corals was measured in a similar manner and calculated as the ratio between the integrated incident irradiance, measured as the reflectance from the reflective standard (350–500 nm) and the integrated coral fluorescence emission (500–600 nm) following Mazel[76].

**Ex situ attraction experiment**. A 20 cm × 7 cm × 7 cm custom-made acrylic chamber was used for the ex situ attraction experiments. The chamber was colored white and filled with 4 L seawater. Targets comprising Petri dishes (9 cm diameter) were painted with either a fluorescent dye (Rust Oleum spray "fluorescent green" or "fluorescent orange"), a non-fluorescent dye (DecoArt acrylic paint "Bright Orange" or "Grasshopper Green"), a white dye coated with a neutral density filter (for the reflective target), or left clear (for the control; see Supplementary Table 1 for spectral

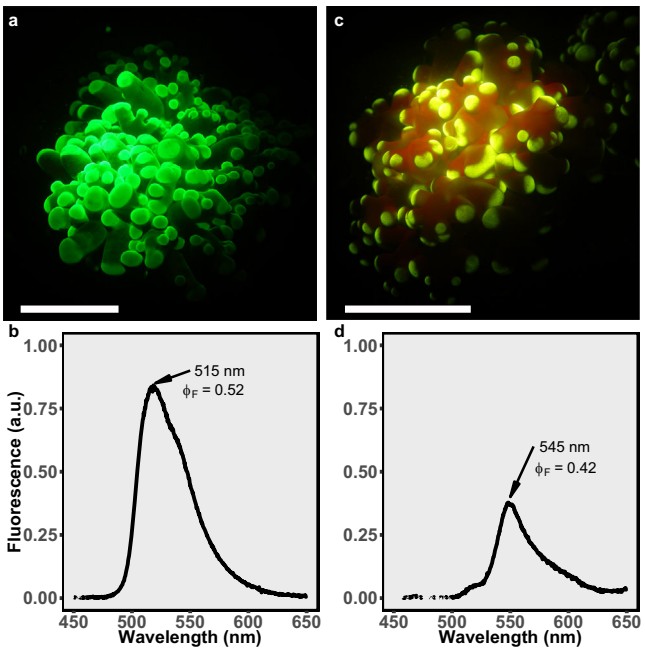

**Fig. 5 Fluorescence morphs of the mesophotic coral *Euphyllia paradivisa*.** **a**, **c** Fluorescent images and corresponding **b**, **d** fluorescence emission spectra of **a**, **b** green and **c**, **d** yellow fluorescence morphs of *E. paradivisa*. The relative coral fluorescence emission peaks (515 nm for the green morph and 545 nm for the yellow morph) and fluorescence efficiency ($\Phi_F$) are indicated by arrows. Scale bars indicate 1 cm.

information). All dishes were cast in clear epoxy in order to avoid any differences in chemical signaling from the dyes. In each trial, two targets were placed on opposite sides of the chamber. Adult *Artemia salina* were chosen for this experiment due to their size, accessibility, and ease of maintenance. They are reported to have an absorbance peak at 520 nm which is intensity dependent[51]. *A. salina* individuals ($n = 22$–63 individuals per trial, size ~6 mm) were released into the chamber and given 15 min to swim. The chamber was then separated into two equal sections by a plastic divider and the number of *A. salina* in each section of the chamber was counted. Prior to each trial, the seawater was replaced and the position of the targets was switched in order to avoid any bias towards one of the sides. The experiment was conducted under blue light illumination (Supplementary Fig. 1; OSRAM L63W/67 blue bulbs, ~30 µmol photons m$^{-2}$ s$^{-1}$) and repeated ten times for each combination. It was then repeated with freshly caught mysids (*Anisomysis marisrubri*, size ~7 mm, $n = 10$ for each combination) from the shallow waters near the Inter-university Institute for Marine Sciences in Eilat (IUI) and with 5-day-old mariculture sea fish larvae (*Sparus aurata*, size ~6 mm, $n = 5$ for each combination). The latter observations were performed at the mariculture center in which the fish are reared, by a certified personal. No special permits were required and the larvae were not harmed or sacrificed while performing the observations.

**In situ attraction experiment**. Three custom-made plankton traps were specifically built for this experiment. Each trap had a volume of 1 L and twelve 25 mm diameter funnel-like entrance points, each with a 5 mm hole at its tip. Inside each trap was a clear (control) or fluorescent (green or orange) target placed inside a clear acrylic glass cylinder in order to prevent any chemical signaling from the dyes (see Supplementary Table 1 for spectral information). The traps were covered with a 40 µm mesh and mounted on a triangular metal frame at equal distances from one another (90 cm apart). The traps were carefully positioned by divers on the sea bottom at 40 m depth near the reef in front of the IUI, and the mesh was then removed. Each trial was initiated ~3 h after sunrise and the traps were left in the sea for four hours at each trial. The experiment was repeated six times, each time changing the orientation of the frame. During retrieval of the traps and until the samples were analyzed, the traps were covered by a 40 µm mesh in order to prevent the loss of plankton. Following the retrieval of the traps, and within 4 h (during which the traps were kept on ice), their contents were filtered through a 40 µm mesh, separated into size fractions (40–200, 200–500, and >500 µ), and the collected plankton were counted and taxonomically identified under a dissecting microscope.

**Ex situ predation experiment**. Four colonies of the scleractinian coral *Euphyllia paradivisa* were collected from the Dekel Beach at Eilat (29°32'17"N, 34°56'56"E) at 45 m depth, during an open-circuit technical dive. Approximately five polyps of similar size from each colony were detached underwater using a bone cutter and placed in a Ziploc bag. Samples were transferred to the IUI running seawater system and kept under a 12:12 h light-dark regime supplied by four OSRAM L63W/67 blue light bulbs to create light conditions similar to those at the collection site (Supplementary Fig. 1). Corals were defined as green or yellow fluorescence morphs ($n = 2$ colonies from each morph) according to their dominant fluorescence emission peak (Supplementary Table 1). The diameter of each polyp was measured with a caliper and the circular surface area of the coral was calculated. This parameter was used for normalization since the coral tentacle surface area rapidly changes while the coral is feeding (expands and contracts), whereas the skeleton-based measurement remains constant. Individual detached *E. paradivisa*

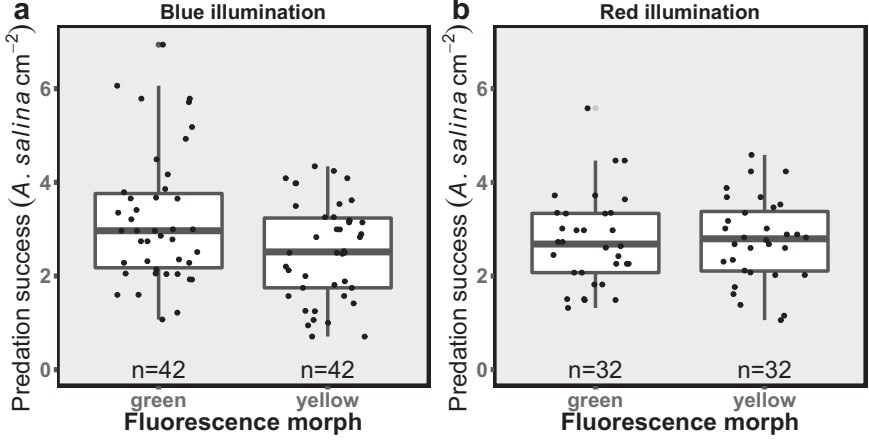

**Fig. 6 The predation success of green and yellow fluorescence morphs of *Euphyllia paradivisa*.** **a** Predation success (number of captured *Artemia salina* normalized to cm$^2$) of green and yellow fluorescence morphs of the coral *E. paradivisa* under blue illumination that excites fluorescence; and **b** under fluorescence non-exciting red illumination. The number of trials ($n$) for each morph is indicated below each box. Boxes represent the upper and lower quartile, centerlines represent medians, and whiskers extend to data measurements that are less than 1.5*IQR away from the first/third quartile. Gray circle represents an outlier.

polyps were each placed in 2 L black containers filled with fresh seawater in each trial. Corals were given 10 min to acclimate to the transfer with running seawater in order to wash away any secreted mucus. After a water level adjustment (to maintain an equal volume of water in each trial), this incubation was followed by a 10 min incubation without water flow to allow the contracted tentacles to re-expand. Following the acclimation period (total of 20 min), 20 A. salina individuals were introduced into the container and the corals were given 30 min to feed, a period that we had found to be optimal. Following the 30 min feeding period, corals were removed from the container to ensure that feeding had ceased. The remaining, swimming, A. salina were collected from the containers and counted. The feeding success was calculated by dividing the number of predated A. Salina by the calculated size of the polyp. This experiment ($n = 42$ for each morph) was performed under blue light illumination, which excites the natural coral fluorescence (Supplementary Fig. 1), and then repeated ($n = 32$ for each morph) under red light illumination (PHILIPS TL-D 36 W/15 red bulbs), which does not excite the fluorescence (Supplementary Fig. 1).

**Statistics and reproducibility**. All data were statistically analyzed using R software[77]. Generalized linear mixed-effects models (GLMM) using a binomial distribution (for all ex situ attraction analyses), normal distribution (for the in situ attraction analyses), or gamma/normal distribution (for predation experiments analyses) were constructed using "lme4"[78]. For the ex situ attraction experiment, models were constructed with the color of the target considered as a fixed effect and the trial number and side of the chamber as random effects. For the in situ attraction experiment, we considered the color of the trap as a fixed effect and the jar, position on the frame, and trial number as random effects. For the ex situ predation experiment, we considered the color of the coral as a fixed effect and the colony nested within color as a random effect. The best model was chosen based on the Akaike information criterion (AIC) score and the selected model was tested with ANOVA. Cohen's d was calculated using "EMAtools"[79]. The model residuals were tested for normality, visually and with Shapiro–Wilk test, and for homogeneity of variance, visually and with Leven's test. All data and models passed the required assumptions. See Supplementary Table 2 for detailed statistical information.

**Reporting summary**. Further information on research design is available in the Nature Research Reporting Summary linked to this article.

## Data availability

The datasets generated and/or analyzed during the current study are available at https://figshare.com/projects/Coral_fluorescence_a_prey-lure_in_deep_habitats_CB/133898.

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

## Acknowledgements

We would like to thank the Interuniversity Institute for Marine Sciences in Eilat for making their facilities available to us. We thank G. Eviatar for her help in conducting the ex situ predation experiment, L. levy and S. Ayalon for their help with the ex situ attraction experiments; R. Tamir, A. Oren, D. Shefi, G. Eviatar, R. Liberman, L. Levy, D. Churilov, G. Zaltzman, and O. Hameiri for their diving assistance; and all the YL lab members for their support. We thank M. Ohavia for the technical support; B.V. Farstey for plankton identification assistance; and N. Paz for her linguistic editing. This research was funded by the Israel Science Foundation (ISF): ISF-NRF (The National Research Foundation of Singapore) joint research program grant No. 2654/17 to Y.L. and by the Ministry of Science, Technology & Space fellowship agreement No. 3-18487 to O.B.-Z.

## Author contributions

O.B.-Z. and Y.L. designed the experiments, Y.Li assisted in the design and execution of the in situ attraction experiment, and G.E. provided the corals for the ex situ predation experiment. O.B.-Z. analyzed and visualized the data and wrote the first draft. All authors edited and approved the final version of this manuscript.

## Competing interests
The authors declare no competing interests.
