## [Peer Review File · Communications Biology]

Reviewers' comments:

Reviewer #1 (Remarks to the Author):

Dear authors, I appreciate your efforts to fulfill the research gaps in the marine fluorescence phenomena. It is an interesting piece of work. I have included a few comments for improving the discussion section.

Line 17: replace "its biological role in"... with "its biological role is"

Line 152: is it orange or red?? Check figure 4A. correct it throughout the manuscript.

Line 261-263: what grade or manufactured dyes have been used. Kindly provide the details.

Line 311: previous preliminary experiments to be optimal.. add the reference for this study.

Line 314: it is not clear to me whether or not the authors have kept the various calculated sizes of the polyps in the same tank or different tanks to check feeding success?

Authors shall include a small section on "visual characteristics and cones of the selected experimental organisms". Include the discussion part with this information, from line 196 onwards. Shallow water corals also possess fluorescence traits and display prey attraction ability. So, how do authors justify the difference between the Shallowater and MCE corals, with respect to prey attraction via fluorescence? What about chemical signaling between coral and plankton???

Add these references and improve the discussion:

<https://royalsocietypublishing.org/doi/10.1098/rspb.2009.1774>

<https://royalsocietypublishing.org/doi/10.1098/rspb.2015.1055>

<https://www.sciencedirect.com/science/article/abs/pii/S2352485518305954>

Best wishes

Reviewer #2 (Remarks to the Author):

The authors report evidence supporting a new function (prey attraction) for green fluorescence in reef building corals. The role of green fluorescent proteins (GFP) host pigments has confounded investigators for decades. While several potential functions/roles have been suggested and claimed over the years, solid evidence for most (except photoprotection) has been elusive. Unfortunately, contradictory evidence for many of these purported roles is often ignored by contemporary coral reef scientists in their publications.

The authors conduct multiple, laboratory and in situ experiments that are very well designed to test their hypothesis. The data and analyses are robust, firmly support their hypothesis, and deserves to be published/shared with the larger scientific community. Their findings confirm a new function for the role of GFPs which is of interest to the wider marine ecology community.

The introduction presents a background review of what is known about the role of GFPs in corals. However, it is somewhat selective/incomplete in the citations which can be potentially misleading to the uninformed reader. In the humble opinion of this one reviewer, the authors could be less selective and more holistic to cite the most relevant and most important conflicting studies and/or reviews in a more balance manner to avoid unintentionally misleading the reader. It may generate a less compelling story to tell, but it will be closer to the truth. Science is sometimes messy with no shortage of inconvenient details.

- A number of roles for coral GFPs are listed. Two are explained in more detail. One (photosynthesis enhancement) is supported. One (photoprotection) is contradicted. Has the contemporary knowledge to date been fairly and accurately represented? In both cases supporting citations are used, but they are selective. Contradicting citations are not acknowledged.
- For example, the role of photoprotection has been experimentally well supported in shallow water corals by multiple studies in detail but not cited/acknowledged (e.g., papers from J. Wiedenmann's lab). As written, a less informed reader is given the impression that this role has been debunked since this role tested negative in mesophotic corals.
- Other examples (e.g., photosynthesis enhancement) are noted in the detailed comments
- If these omissions stem from the publisher's limit on number of citations, then the authors are unfairly constrained, and consequences are not in the best interest of science.

In the discussion, the authors establish the strong evidence for their prey attraction hypothesis but stop short of explaining/speculating why green wavelengths (same color as phytoplankton) attract prey and other wavelengths do not; and why some prey (herbivores) are attracted while other prey (carnivores) are not. I have included some suggestions in the detailed comments that might help fill this gap from an ecological perspective. Perhaps they are too speculative, but the authors can decide if it adds value. Incidentally, this same hypothesis was first posited for an azooxanthellate coral in an unpublished dissertation (Kahng 2006).

In general, the manuscript is well written, but the Discussion section should be structured into multiple paragraphs to aid readability and focus the reader on the major points.

Detailed Comments:

Title

This function is probably not exclusive to deep water, but to low light environments in general which could include shaded habitat, certain times of the day, and/or certain atmospheric conditions

Line 20-22

I do not believe there is sufficient evidence to generalize that PS is not the main energy source (organic C) for all mesophotic corals. While this line of thought has become dogma and a default assumption, the supporting evidence is far from conclusive to generalize. There are certainly multiple studies including older ones that support this conclusion for multiple species. However, there also more recent studies that directly contradict the very foundation (e.g., the utility/accuracy of bulk stable isotopes methods in assessing trophic status for zooxanthellate corals) of some of these older studies (e.g., Wall et al. 2019;) and/or directly contradict their conclusions for the same species (e.g., Lesser et al. 2010 vs Crandal et al. 2016; Mass et al. 2007 vs Martinez et al. 2020).

Studies applying older methods must be interpreted in light of recent advances in technology and knowledge (and not at face value). Generalizing across all zooxanthellate scleractinian corals should be done with extreme caution given their polyphyletic lineages that predate the fossil record into the Cambrian period (McFadden et al. 2021). It is quite common for different coral species to employ different, sometimes opposing strategies with respect to photosynthesis and heterotrophy (reviewed in Houlbreque Ferrier-Pages 2009; Kahng et al. 2019).

It is also important to note that not all older studies have been critically reviewed in the published literature in light of accelerating contemporary knowledge and explosion of primary literature.

- For example, the often cited in situ respirometry chambers study by Fricke et al. (1987) placed the corals (*Leptoseris fragilis*) with epibionts inside small plexiglass chamber at depth for 30-60 min with the light sensors outside the chambers. This design introduces small artifacts (perhaps unavoidably) that systematically overestimates the light energy available to the corals, underestimates coral oxygen production, and dampens hydrodynamic flow and mass transfer which slow rates of PS (e.g., Mass et al. 2010; reviewed by Lowe and Falter 2015). Cumulatively, these artifacts introduce uncertainty surrounding the original quantitative conclusion.

Line 43-45

These citations are highly selective, misleading, and not representative of what is known about this hypothesis. Beyond the early circumstantial evidence presented in the 1980's, all empirical evidence and theoretical scrutiny since then does not support any such role (e.g., Mazel et al. 2003; Gilmore et al. 2003; etc.). It has been viewed multiple times by multiple reputable scientists (Lesser et al. 2004; Dubinsky and Falkowski 2011; Kahng et al. 2019) and the scientific consensus remains. Yet, this purported function continues to be repeatedly propagated in the contemporary literature (perhaps because it sounds sensational to the uninformed?) despite the overwhelming and convincing evidence against it over the past two decades.

Line 56-57

Is there a reason for not acknowledging conflicting evidence using more detailed/direct/reliable methodologies on trophic status? e.g., Crandall et al. (2016), Martinez et al. (2020), etc.

Line 59-60

There is abundant zooplankton literature about differential availability of zooplankton at night vs the day when they hide from visual planktivores. Feeding behavior is stimulated by the presence of food in many organisms across many taxa. While radiation-induced damage is surely part of the story, the reader is left with the impression that it is the only factor driving this behavior. Is this accurate?

Line 61

The dominant Indo-Pacific zooxanthellate coral taxon (e.g., *Leptoseris* spp.) in the lower photic zone completely lack tentacles (Kahng et al. 2020). Yet some of these colonies haphazardly exhibit GFPs, sometimes in radial patterns (not sure whether this common-knowledge observation was actually ever published anywhere).

Line 70-92

excellent experiments!

Several species of copepods also exhibit differential spectral phototaxis with peak response in the vicinity of peak emissions for some GFPs (~520nm) but outside the peak for downwelling light (~410-480 nm) in oligotrophic mesophotic habitat (Cohen and Forward 2002). While these particular species copepods were reacting to twilight conditions in coastal water as a DVM signal, it demonstrates the ability of zooplankton visual systems to discriminate and select specific wavelengths for phototaxis.

Line 195-246

Single long paragraph should be structured into organized set of major points with one paragraph for each

Content wise, very good treatment of the topic but, . . .

Why green?!?! To strengthen the discussion, maybe add a few lines of text on the green reflectance (low absorption) from chlorophyll (phytoplankton) as a byproduct of PS. Phototaxis by herbivorous zooplankton towards their food source would offer a selective advantage. Your *Sparus aurata* fish larvae feed on zooplankton and not phytoplankton, so their lack of green phototaxis is consistent with food source. In contrast, *Artemia salina* is a filter feeder of phytoplankton cells. It appears that your light trap data is consistent as a majority of zooplankton biomass is herbivorous. However, predators (e.g., chaetognaths) attracted to such herbivorous prey (chemical cues, mechanical detection, visual detection, etc.) would also be captured (scientific bycatch).

See unpublished text in Kahng (2006) dissertation on the potential role of GFPs in an azooxanthellate octocoral *Carijoa riisei* (p.77-82). The same GFP prey attraction hypothesis is proposed and discussed relative to the available evidence with a similar conclusion (however, based more on deductive reasoning than experimental data).

Response to reviewers-Communications biology

Referee expertise:

Referee #1: fluorescence, coral reefs, marine molecular biology

Referee #2: coral reef ecology

Reviewers' comments:

Reviewer #1 (Remarks to the Author):

Dear authors, I appreciate your efforts to fulfill the research gaps in the marine fluorescence phenomena. It is an interesting piece of work. I have included a few comments for improving the discussion section.

- 1.1 Line 17: replace "its biological role in"... with "its biological role is"
- 1.1 "in" replaced with "is" (line 19 in the revised manuscript).

- 1.2 Line 152: is it orange or red?? Check figure 4A. correct it throughout the manuscript.
- 1.2 The traps were named according to their fluorescence emission peak and not their appearance. Therefore, while the fluorescent trap in Fig. 4A might look red, its emission peak (595 nm, see Table S1) indicates that it is in the orange portion of the spectrum. The trap color in the x-axis of Fig. 4B has been corrected from red to orange. To prevent any confusion, the colors of the boxes in Figures 1-3 have been revised (as well as the caption) and the previously red color has now been replaced with shades of orange (see figures 1-3 in the revised manuscript).

- 1.3 Line 261-263: what grade or manufactured dyes have been used. Kindly provide the details.
- 1.3 The targets were colored using a commercial fluorescent paint spray (Rust Oleum "fluorescent green" and "fluorescent orange" or acrylic non-fluorescent dyes DecoArt "bright orange" and "grasshopper green"). This information has been added to the Methods section, lines 316-317 in the revised manuscript.

- 1.4 Line 311: previous preliminary experiments to be optimal.. add the reference for this study.
- 1.4 The preliminary experiments were performed by us, during this study. We tested different incubation times and found that 30 min was optimal, as longer incubations did not increase the predation rate.

- 1.5 Line 314: it is not clear to me whether or not the authors have kept the various calculated sizes of the polyps in the same tank or different tanks to check feeding success?
- 1.5 Each polyp was tested individually. Both tank size and water volume were kept constant, thereby maintaining the same *Artemia* concentration among trials. Additionally, the use of large tanks in comparison to the size of the polyps, reduced the effect of coral volume/size on prey concentration. See revised text in lines 368-369 in the revised manuscript (“After a water level adjustment (to maintain an equal volume of water in each trial), this incubation was followed...”).
- 1.6 Authors shall include a small section on “visual characteristics and cones of the selected experimental organisms”. Include the discussion part with this information, from line 196 onwards.
- 1.6 Although the data for some of our studied organisms are limited, a section discussing the visual sensitivities and phototaxis of the selected experimental organisms in this study has now been added (lines 230-237 in the revised manuscript).
- 1.7 Shallow water corals also possess fluorescence traits and display prey attraction ability. So, how do authors justify the difference between the Shallowater and MCE corals, with respect to prey attraction via fluorescence? What about chemical signaling between coral and plankton???
- 1.7 As a first test of the possible role of coral fluorescence as a prey lure, we examined this possibility in an environment that naturally enhances the excitation and visibility of fluorescence, thus providing support for the hypothesis. Fluorescence may indeed be attracting prey in shallower habitats, but this aspect should be further tested in the future, along with other prey attraction mechanisms such as chemical signaling. Our current attraction experiments, however, using fluorescent objects that did not differ in their chemical output, provide evidence that plankton respond only to the fluorescence, regardless of chemical signaling. Another possibility, which is clear from the variety of suggested roles for fluorescence, is that it may serve different or multiple roles under different conditions.

Add these references and improve the discussion:

<https://royalsocietypublishing.org/doi/10.1098/rspb.2009.1774>

<https://royalsocietypublishing.org/doi/10.1098/rspb.2015.1055>

<https://www.sciencedirect.com/science/article/abs/pii/S2352485518305954>

Best wishes

Reviewer #2 (Remarks to the Author):

The authors report evidence supporting a new function (prey attraction) for green fluorescence in reef building corals. The role of green fluorescent

proteins (GFP) host pigments has confounded investigators for decades. While several potential functions/roles have been suggested and claimed over the years, solid evidence for most (except photoprotection) has been elusive. Unfortunately, contradictory evidence for many of these purported roles is often ignored by contemporary coral reef scientists in their publications.

The authors conduct multiple, laboratory and in situ experiments that are very well designed to test their hypothesis. The data and analyses are robust, firmly support their hypothesis, and deserves to be published/shared with the larger scientific community. Their findings confirm a new function for the role of GFPs which is of interest to the wider marine ecology community.

Thank you for your supportive comments. We are delighted that you find both the subject and the current study interesting.

- 2.1 The introduction presents a background review of what is known about the role of GFPs in corals. However, it is somewhat selective/incomplete in the citations which can be potentially misleading to the uninformed reader. In the humble opinion of this one reviewer, the authors could be less selective and more holistic to cite the most relevant and most important conflicting studies and/or reviews in a more balance manner to avoid unintentionally misleading the reader. It may generate a less compelling story to tell, but it will be closer to the truth. Science is sometimes messy with no shortage of inconvenient details.
- 2.1 We understand the referee's point. We have focused on some of the main roles suggested for coral fluorescence, especially those suggested for mesophotic environments as the study was aimed at this ecosystem. However, we have now added more citations, some supporting the suggested roles and others contradicting them, to provide the reader with a more comprehensive picture. See lines 38-58 in the revised manuscript.
- 2.2 A number of roles for coral GFPs are listed. Two are explained in more detail. One (photosynthesis enhancement) is supported. One (photoprotection) is contradicted. Has the contemporary knowledge to date been fairly and accurately represented? In both cases supporting citations are used, but they are selective. Contradicting citations are not acknowledged.
- 2.2 We would argue that both hypotheses are controversial. The (leading) photoprotection hypothesis was supported previously by the work of Salih et al. 2000, Smith et al. 2013, Bollati et al. 2020, and other. However, this role may be more complicated if we consider the optical loop created by FPs or may not hold in deeper habitats as previously shown by our group (Ben-Zvi et al. 2019). The same applies to the photosynthesis enhancement role which was logically suggested and partially supported by Schlichter et al. 1986, and further explored and modified by Smith et al. 2017 but failed to be proven for mesophotic corals (Ben-Zvi et al. 2021 and Roth et al. 2015). Many of the above citations are indeed included in the manuscript but they appear in different parts of the text. We have now also included more references (some are contradicting) in lines 50-51 in the revised manuscript.

2.3 For example, the role of photoprotection has been experimentally well supported in shallow water corals by multiple studies in detail but not cited/acknowledged (e.g., papers from J. Wiedenmann's lab). As written, a less informed reader is given the impression that this role has been debunked since this role tested negative in mesophotic corals.

2.3 The literature review of the suggested roles for coral fluorescence has been expanded to include more aspects of each role in lines 41-45 of the revised manuscript.

- Other examples (e.g., photosynthesis enhancement) are noted in the detailed comments
- If these omissions stem from the publisher's limit on number of citations, then the authors are unfairly constrained, and consequences are not in the best interest of science.

In the discussion, the authors establish the strong evidence for their prey attraction hypothesis but stop short of explaining/speculating why green wavelengths (same color as phytoplankton) attract prey and other wavelengths do not; and why some prey (herbivores) are attracted while other prey (carnivores) are not. I have included some suggestions in the detailed comments that might help fill this gap from an ecological perspective. Perhaps they are too speculative, but the authors can decide if it adds value. Incidentally, this same hypothesis was first posited for an azooxanthellate coral in an unpublished dissertation (Kahng 2006).

In general, the manuscript is well written, but the Discussion section should be structured into multiple paragraphs to aid readability and focus the reader on the major points.

Detailed Comments:

2.4 Title

This function is probably not exclusive to deep water, but to low light environments in general which could include shaded habitat, certain times of the day, and/or certain atmospheric conditions

2.4 We believe that the enhanced fluorescent signal is caused by the combination of lower light and bluer spectrum. However, we do not rule out the possibility that this hypothesis is valid for shallower environments or low-light environment. As the current manuscript provides the first examination of a prey attraction role for fluorescence, we targeted the mesophotic environment as we believe it provides an optimal setting for such a role. We hope that this hypothesis will be tested in the future also in other environments, enabling our conclusions to be broadened.

2.5 Line 20-22

I do not believe there is sufficient evidence to generalize that PS is not the main energy source (organic C) for all mesophotic corals. While this line of

thought has become dogma and a default assumption, the supporting evidence is far from conclusive to generalize. There are certainly multiple studies including older ones that support this conclusion for multiple species. However, there also more recent studies that directly contradict the very foundation (e.g., the utility/accuracy of bulk stable isotopes methods in assessing trophic status for zooxanthellate corals) of some of these older studies (e.g., Wall et al. 2019;) and/or directly contradict their conclusions for the same species (e.g., Lesser et al. 2010 vs Crandal et al. 2016; Mass et al. 2007 vs Martinez et al. 2020).

- 2.5 The generalized statement regarding the trophic state of corals has been moderated and the Introduction has been revised to include more details on the nutritional sources of corals. See lines 59-70 in the revised manuscript.
- 2.6 Studies applying older methods must be interpreted in light of recent advances in technology and knowledge (and not at face value). Generalizing across all zooxanthellate scleractinian corals should be done with extreme caution given their polyphyletic lineages that predate the fossil record into the Cambrian period (McFadden et al. 2021). It is quite common for different coral species to employ different, sometimes opposing strategies with respect to photosynthesis and heterotrophy (reviewed in Houlbreque Ferrier-Pages 2009; Kahng et al. 2019).

It is also important to note that not all older studies have been critically reviewed in the published literature in light of accelerating contemporary knowledge and explosion of primary literature.

- For example, the often cited in situ respirometry chambers study by Fricke et al. (1987) placed the corals (*Leptoseris fragilis*) with epibionts inside small plexiglass chamber at depth for 30-60 min with the light sensors outside the chambers. This design introduces small artifacts (perhaps unavoidably) that systematically overestimates the light energy available to the corals, underestimates coral oxygen production, and dampens hydrodynamic flow and mass transfer which slow rates of PS (e.g., Mass et al. 2010; reviewed by Lowe and Falter 2015). Cumulatively, these artifacts introduce uncertainty surrounding the original quantitative conclusion.

- 2.6 As the reviewer stated, this subject is under debate and heavily depends on the methods used in the examination and their analysis. We agree that the common dogma may not apply for all mesophotic corals (or for all corals for that matter). We have now revised this section and further discuss the current knowledge on coral nutrition in general and in MCEs specifically in lines 64-70 in the revised manuscript.

- 2.7 Line 43-45
These citations are highly selective, misleading, and not representative of what is known about this hypothesis. Beyond the early circumstantial evidence presented in the 1980's, all empirical evidence and theoretical scrutiny since then does not support any such role (e.g., Mazel et al. 2003; Gilmore et al. 2003; etc.). It has been viewed multiple times by multiple reputable scientists (Lesser et al. 2004; Dubinsky and Falkowski 2011; Kahng et al. 2019) and the scientific consensus remains. Yet, this purported function

continues to be repeatedly propagated in the contemporary literature (perhaps because it sounds sensational to the uninformed?) despite the overwhelming and convincing evidence against it over the past two decades.

- 2.7 As an introduction, the hypothesis is introduced as a possible and relevant role of coral fluorescence. The photosynthesis enhancement role is also further discussed and somewhat rejected later in the text, therefore presenting a chronologic evolution of this hypothesis. In early studies, such as these cited in lines 39-40 of the revised manuscript, fluorescence was suggested to enhance photosynthesis specifically in MCEs, by converting short, inappropriate wavelengths to more photosynthesis-suitable ones. However, this was later tested and mostly not supported (citations in line 51 in the revised manuscript). In the Introduction we merely present the possible roles for coral fluorescence and then move to the testing of these roles in shallow and mesophotic corals. There is some evidence (circumstantial and perhaps not as convincing as one might hope) for this role in Smith et al. 2017 and Wangpraseurt et al. 2019.
- 2.8 Line 56-57
- Is there a reason for not acknowledging conflicting evidence using more detailed/direct/reliable methodologies on trophic status? e.g., Crandall et al. (2016), Martinez et al. (2020), etc.
- 2.8 We have revised this section of the Introduction (lines 67-70 in the revised manuscript) to include additional updated knowledge, as well as contradictory evidence regarding coral nutrition and the work done in this field. We thank the referee for his comment and suggested citations.
- 2.9 Line 59-60
- There is abundant zooplankton literature about differential availability of zooplankton at night vs the day when they hide from visual planktivores. Feeding behavior is stimulated by the presence of food in many organisms across many taxa. While radiation-induced damage is surely part of the story, the reader is left with the impression that it is the only factor driving this behavior. Is this accurate?
- 2.9 Hypothetically, corals should keep their tentacles expanded during the day in order to maximize their nutritional options, as by doing so they can both photosynthesize while capturing occasional prey. Following the referee's comment, we have rephrased this section and now include other possible reasons for tentacle retraction "...only expand them during the night⁴⁴ in order to avoid radiation-induced damage⁴⁵ and depending on other factors (i.e. flow rate, photosymbionts density, oxygen concentration, and presence of food)⁴⁴⁻⁴⁶..." (see lines 72-76 in the revised manuscript).
- 2.10 Line 61
- The dominant Indo-Pacific zooxanthellate coral taxon (e.g., *Leptoseris* spp.) in the lower photic zone completely lack tentacles (Kahng et al. 2020). Yet some of these colonies haphazardly exhibit GFPs, sometimes in radial

patterns (not sure whether this common-knowledge observation was actually ever published anywhere).

2.10 We do have some experience with this genus, and they indeed lack tentacles and some of the species are actually brightly fluorescence. However, as previously suggested, fluorescence may have multiple roles and in *Leptoseria* we suspect that it serves a light mediation function (or even merely being a byproduct of other processes in the coral host).

2.11 Line 70-92

excellent experiments!

Several species of copepods also exhibit differential spectral phototaxis with peak response in the vicinity of peak emissions for some GFPs (~520nm) but outside the peak for downwelling light (~410-480 nm) in oligotrophic mesophotic habitat (Cohen and Forward 2002). While these particular species copepods were reacting to twilight conditions in coastal water as a DVM signal, it demonstrates the ability of zooplankton visual systems to discriminate and select specific wavelengths for phototaxis.

2.11 We thank the referee for the compliment. Although this response may also explain the copepods swimming into our green traps as they are attracted to the twilight wavelengths, we feel this explanation nonetheless to be somewhat speculative.

2.12 Line 195-246

Single long paragraph should be structured into organized set of major points with one paragraph for each

Content wise, very good treatment of the topic but, . . .

Why green?!?! To strengthen the discussion, maybe add a few lines of text on the green reflectance (low absorption) from chlorophyll (phytoplankton) as a byproduct of PS. Phototaxis by herbivorous zooplankton towards their food source would offer a selective advantage. Your *Sparus aurata* fish larvae feed on zooplankton and not phytoplankton, so their lack of green phototaxis is consistent with food source. In contrast, *Artemia salina* is a filter feeder of phytoplankton cells. It appears that your light trap data is consistent as a majority of zooplankton biomass is herbivorous. However, predators (e.g., chaetognaths) attracted to such herbivorous prey (chemical cues, mechanical detection, visual detection, etc.) would also be captured (scientific bycatch).

2.12 This point (the “why green?” question) was addressed in lines 201-205 of the Discussion in the original manuscript, referring to herbivores’ known response to the color of chlorophyll (lines 244-246 in the revised manuscript). We have followed the referee’s suggestion and broadened this discussion with a targeted section for each of our tested organisms (for the *ex-situ* experiments) and the natural plankton assemblages in lines 230-237 in the revised manuscript).

2.13 See unpublished text in Kahng (2006) dissertation on the potential role of GFPs in an azooxanthellate octocoral *Carijoa riisei* (p.77-82). The same GFP prey attraction hypothesis is proposed and discussed relative to the available

evidence with a similar conclusion (however, based more on deductive reasoning than experimental data).

- 2.13 We were unaware of the section about fluorescence in Kahng's dissertation but are familiar with the Reef Site from 2005, where the function of FPs in *C. riisei* was presumed to be photoprotective. Our current thinking regarding the reason behind the attraction of plankton to green fluorescence would seem to be similar.

REVIEWERS' COMMENTS:

Reviewer #1 (Remarks to the Author):

The authors have revised the manuscript and it looks perfect to me. This can be accepted in your esteemed journal.

thanks for the opportunity given to me to review this article.

Reviewer #2 (Remarks to the Author):

Excellent study. My sincere apologies for missing the Smith et al (1953) reference during my original review. I was too hasty in my comments. The revisions provide a strong, comprehensive/balanced assessment of the science surrounding phenomena reported.

I disagree with the request from reviewer #1 to add the three additional GFP references on Ctenophora and coral monitoring. There are numerous references (>>100) on the topic of GFPs in marine organisms, and the authors have thoughtfully cited the most relevant with respect to potential GFP function in corals. Adding select and blatantly irrelevant citations would dilute the focused nature of this report. Such requests require specific scientific justification and should be free from any professional conflict of interest.